# Physical Activity, Life Satisfaction, Stress Perception and Coping Strategies of University Students in Belarus during the COVID-19 Pandemic

**DOI:** 10.3390/ijerph19148629

**Published:** 2022-07-15

**Authors:** Andrei Shpakou, Ihar A. Naumau, Tatyana Yu. Krestyaninova, Alena V. Znatnova, Svetlana V. Lollini, Sergei Surkov, Aleh Kuzniatsou

**Affiliations:** 1Department of Integrated Medical Care, Faculty of Health Sciences, Medical University of Bialystok, 15-089 Białystok, Poland; 2Department of General Hygiene and Ecology, Grodno State Medical University, 230009 Grodno, Belarus; kge_grgmu@mail.ru; 3Department of Psychology, Vitebsk State University Named after P.M. Masherov, 210038 Vitebsk, Belarus; auta@bk.ru; 4Department of Physical Education and Sports, Belarusian State Pedagogical University Named after Maxim Tank, 220030 Minsk, Belarus; lena.znatnova2010@gmail.com; 5Department of Ecological and Preventive Medicine, Vitebsk State Order of Peoples’ Friendship Medical University, 210009 Vitebsk, Belarus; svlollini@gmail.com; 6Faculty of Physical Education and Sports, Brest State A.S. Pushkin University, 224016 Brest, Belarus; surkov.brsu@gmail.com; 7Institute of Biochemistry of Biologically Active Compounds of the National Academy of Sciences of Belarus, 230023 Grodno, Belarus; olegkuznetsov@inbox.ru

**Keywords:** Belarus, students, physical activity, life satisfaction, stress, coping strategies, COVID-19

## Abstract

The COVID-19 restrictions in Belarus turned out to be less stringent than those of its neighboring countries. Objective: We aimed to evaluate physical activity, life satisfaction, perception of stress, choice of coping strategies and their correlations among Belarusian students. An anonymous Internet survey was conducted among 1769 students studying at the faculties of physical culture (415), pedagogical (737), and medical (617) universities. International Physical Activity Questionnaires (IPAQ) and Satisfaction With Life Scale (SWLS), Perceived Stress Scale-10 (PSS-10), and Coping Orientations to Problems Experienced (mini-COPE) questionnaires were used. The respondents declared sufficient and high levels of physical activity. The level of physical activity was correlated with life satisfaction (typically for student-athletes). The least satisfaction with life and highest level of perceived stress were among future doctors. The minimum indicator of stress was noted in athletes. The most common coping strategy was active coping. Strategies of problem avoidance and seeking support from outside were not used by student-athletes. These strategies were used by pedagogical and medical students. Student-athletes have the most favorable opportunities, followed by representatives of pedagogical and medical students. The proposed classification of the levels of behavior (optimal, acceptable, satisfactory and risky) makes it possible to adjust lifestyles.

## 1. Introduction

The rapid spread of COVID-19 has changed the lives of millions of people around the world in a short time [1]. Significant changes in lifestyle, combined with the risk of infection, the risk of getting sick, anxiety about the loss of social contacts, and concern about the health of relatives can be considered as risk factors for maladjustment in a pandemic [2]. In order to prevent the expansion of the SARS-CoV-2 virus, countries have taken various restrictive measures [3,4,5]. In most European countries, quarantine measures turned out to be quite strict (bans on movement and access to services, a transition to remote education, and other measures typical of a lockdown) [6]. In Belarus, unlike other countries with restrictive anti-pandemic measures, strict quarantine measures have been abandoned [7,8]. It was recommended in the country to maintain the previous organization of social life without serious restrictions on movement, the regime of blocking social contacts was not introduced, and the population was persistently informed about the need to comply with security measures, which was supposed to minimize panic, ease anxiety and reduce the psychological burden in society [9]. In educational institutions, the transition to hybrid learning was recommended, that is, the traditional mode in combination with the use of information and communication technologies [10]. Of all European countries, a similar situation was typical only for Sweden [11], which carried out similar activities based on the extensive advertising of prevention and the offer of voluntary vaccination. The restrictions imposed in Belarus in connection with the COVID-19 pandemic turned out to be much less pronounced in comparison with the anti-pandemic measures in most European countries [12]. They can nevertheless be considered as contributing to Post-Traumatic Stress Disorder (PTSD) for the population [13,14], among which, in relation to the Belarusian student youth, the most significant restrictions during the pandemic were a change in the usual mode of activity, a partial transition to a remote system of conducting classes, a decrease in physical activity, a decrease in life satisfaction, limited contacts, and the perception of the situation as stressful [15,16].

In the current lifestyle, physical activity (PA) is treated as a means to achieve a better satisfaction of life. Seeking personal wellbeing and life satisfaction can be challenging endeavors even in the best of times. The study found an increase in physical inactivity during the transition from adolescence to adulthood and throughout university studies [17]. Active people might be more susceptible to poorer wellbeing during times of restriction. The current situation and the restrictive measures taken were accompanied by worsening life satisfaction, decreased physical activity, and a high mental load [18]. With recent predictions that the COVID-19 pandemic could negatively affect people’s lifestyles, researchers reported that—for quarantined individuals—time spent engaged in all physical activity and metabolic equivalents of tasks in each physical activity intensity zone decreased significantly during COVID-19. The authors pointed out that physical activity level is positively related to life satisfaction [19]. In addition to the stress associated with the danger of getting sick, the destructive effect of decreased physical activity on the satisfaction gained from life is confirmed. The potential changes in their physical activity, along with their overall physical and psychological wellbeing, may have ultimately influenced their life satisfaction and mental health. Total physical activity energy expenditures were significant predictors for the decrease in mental wellbeing [20]. Although the pandemic has been a global event, it has had a somewhat disproportionate impact on certain groups of people, especially those classified as essential workers, such as healthcare providers and teachers. However, no detailed study has examined the impact of Belarusian university students’ change in physical activity, and satisfaction with life during the COVID-19 pandemic [21]. Despite the high health risks associated with these factors during the pandemic, the role of physical activity in Belarusian students’ wellbeing and life satisfaction remains largely unknown. Therefore, it was interesting to observe the condition of student-athletes, future teachers and doctors. These are the most common and closest specialties according to the classification of “Man-Man” types of occupations [22]. This division is based on distinguishing basic human qualities according to their psychophysiological and psychic significance. This is what the sample of the present study was based on.

In this connection, it should be noted that in recent years, in scientific research, special attention has been paid to the experience of stress and the actions that a person takes in order to cope with extreme events, presenting this as “coping with stress” [23,24]. The concept that stress leads to the disturbance of relations (transactions) between the person and the environment is also popular [25,26], and its consequences are caused by the possibility of overcoming a stressful situation with the help of coping strategies [27]. Under the conditions of an unstable state in society during a pandemic, a transformation and expansion of the repertoire of coping strategies for coping with stress was noted [20]. All of this makes it possible to consider the stressful situation caused by the pandemic not only as a negative component of a lifestyle but also as a challenge, and to consider the possibility of active counteraction through the search for new ways to overcome it [28].

The aim of the study was to evaluate physical activity, life satisfaction, the perception of stress, the choice of coping strategies, and their interactions among full-time students of Belarusian universities. The following aims were pursued: (1) to study the features and relationship of physical activity, life satisfaction, the prevalence of stress, and the perception of a stressful situation among students in Belarus studying medical, pedagogy and physical education specialties during the COVID-19 pandemic; (2) to identify the priority strategies for coping with stress (coping strategies) used by students; and (3) to suggest predictive criteria for the minimization of the stressful consequences of the COVID-19 pandemic, based on a combination of indicators of situational adaptation in terms of physical activity, life satisfaction, and the perception of stress.

An attempt was made to find out whether there were differences in the choice of coping strategies when responding to a stressful situation among student-athletes, future doctors, and representatives of the pedagogical students. It was assumed that the lack of physical activity and the decrease in life satisfaction during a pandemic were due to a high level of stress and the use of non-adaptive coping strategies to overcome it. Such a study was carried out in Belarus for the first time.

## 2. Materials and Methods

### 2.1. Study Design and Population

This study is based on a cross-sectional survey carried out between January and February 2021. This is a fragment of a larger international multi-center research project—The COVID-19 Coping Study of Students from East Europe (SEECoping-S). An invitation to participate in an online survey (Google form) was distributed through targeted advertising, including an e-learning platform (Moodle), Skype, Microsoft Teams, and university social networks. The proposed information resources were available to students and were widely used in teaching during the COVID-19 pandemic. Prior to the start of the study, all of the participants were informed of its objectives, methodology, and the anonymous and confidential nature of the survey. The questionnaire contained information about the study, its goals and objectives; an invitation to participate through informed consent; and socio-demographic data on age, gender, education profile, self-reported physical activity, contact with patients with COVID-19, being in quarantine or strict self-isolation, infection with the SARS-CoV-2 virus, and vaccination course.

The main part of the questionnaire included a set of generally accepted standardized questionnaires in the Russian version, in order to determine the level of declared physical activity, life satisfaction, the assessment of stress experience, and the use of coping strategies.

Permission was obtained from the leadership of the universities participating in the study to conduct an anonymous survey of their students. The research project was approved by the heads of the institutions of higher education where the research was conducted, and by the Bioethical Commission of the Medical University (Bialystok, Poland). All of the procedures carried out with the participation of people were in accordance with ethical standards, as well as the Helsinki Declaration of 1964 (as amended and supplemented).

Anonymous surveys were completed by 1769 full-time students of 2–5 courses from six universities in Belarus. The respondents were from the Faculty of Physical Education in Grodno, Brest and Vitebsk (*n* = 415); Pedagogical faculties in Grodno, Minsk and Vitebsk (*n* = 737); and the Medical University in Grodno and Vitebsk (*n* = 617). In order to reflect the broad representation of students, the nation’s leading regional and central university were chosen for the study. The willingness of leading scholars to participate in the study was also taken into account. The choice of the profile of education was connected with the peculiarities of future labor activity, and readiness to work with people, including in extreme situations.

### 2.2. Study Questionnaire

In the questionnaire, the COVID-19 pandemic and its consequences were considered as the main stressors affecting daily life. The study of the level of physical activity was carried out using an abbreviated version (4 items) [29] of the International Physical Activity Questionnaire (IPAQ) [30]. The classification of declared physical activity was carried out on the basis of an estimate of energy expenditure, calculated by multiplying the frequency during the week of performing intense or moderate physical activity and walking by the duration of each of the three types of activity. The result was expressed in units of metabolic equivalents of work (MET min per week), which is the amount of energy expended by a person at rest, and is equivalent to burning 1 kcal/kg/h, which allows you to correlate the level of metabolism of a person during physical activity to the level of his metabolism at rest. The total physical activity was calculated by summing the MET values obtained from three activities per week. The levels of PA were classified as high (intense PA), sufficient (increased and moderate PA), and low (insufficient PA) [31]. An indicator exceeding 1500 MET-min/week in combination with intensive physical activity for 3 days or more per week or a weekly MET indicator exceeding 3000 was considered as a high level of physical activity. Elevated PA was considered when its values exceeded the indicator of 1500 MET-min/week, but lasting less than 3 days a week. PA was taken as sufficient, amounting to 600–1500 MET-min/week. Insufficient PA was characterized by an indicator of less than 600 MET-min/week or a mismatch of conditions for moderate, high and intense PA.

The Satisfaction With Life Scale (SWLS) proposed by Diener et al. [32] was used to assess cognitive judgments about the subjective perception of wellbeing in life [33]. The respondents indicated the extent to which they agreed or disagreed with each of the five statements in the questionnaire on a 7-point scale (Likert scale). Higher results indicated an increase in the significance of the level of satisfaction with life [34]. Depending on the degree of satisfaction with life, the results were classified as high, medium or low. Cronbach’s (α) reliability analysis was used in order to verify the internal consistency of the questionnaire: the reliability of the tool was assessed as α = 0.915, which is a satisfactory level of reliability.

The Perceived Stress Scale—PSS-10 (Russian version)—was used to determine the level of psychological stress [35,36]. The degree of the subjective perception of life (10 questions) was determined in 5 gradations, ranging from ‘prosperous’ to ‘overloaded with stress’. Initially, the results of the PSS-10 were assessed on subscales: “Overload”, which measures the subjectively perceived level of tension in the situation, and “Stress response”, which determines the level of efforts made to overcome stress. The overall result characterized the degree of perceived stress in gradation from the minimum to the maximum. Regarding internal consistency, the Cronbach alpha value was α = 0.852.

The degree of preference for coping strategies was determined using the mini-COPE (The Coping Orientations to Problems Experienced) questionnaire [37] in a Russian-language adaptation [38]. Coping (14 strategies) was assessed using the shortened version of Brief COPE (28 questions) recommended in 1997 [39]. The respondents, evaluating their choice, noted that it helps to cope with emotions, protect themselves and loved ones, reduce risks and threats, and prepare for the unknown and further developments.

The severity of the scales of coping behavior was classified as the following: (1) active coping (actions to eliminate, reduce the stressor or its consequences), (2) planning (thinking about and planning what to do), (3) positive reframing (thinking about a negative or challenging situation in a more positive way), (4) acceptance (accepting the situation as irreversible, which you need to get used to), (5) humor (as a way to soften unwanted emotions), (6) religion (as a source of emotional support and a pointer to a positive reappraisal), (7) the use of emotional support (sympathy, understanding, moral support), (8) the use of instrumental support (the desire to get advice, help or reliable information), (9) self-distraction (avoiding thoughts about the situation by engaging in other activities), (10) denial (denial of the reality of a stressful situation, ignoring it), (11) venting (focus on emotions and their manifestation, worry about your emotions, a tendency to discharge them), (12) substance use (use of alcohol or other psychoactive drugs), (13) behavioral disengagement (helplessness, submission, refusal of efforts), (14) self-blame. The α-Cronbach coefficient amounted to α = 0.895, and yielded satisfactory results.

The level of coping among the respondents ranged from 0 (no use of this coping strategy) to 3 (the most frequent use) for each coping strategy. All of the responses were grouped into three main coping strategies: active coping (scales 1–3), coping by avoidance (4, 5, 9, 10, 12, 13) and problem-oriented coping (6–8, 11, 14) [40].

### 2.3. Statistical Analysis

The data were processed using the programs of the Statistica 13.0 PL package. In order to describe and generalize the results, and to determine the reliability of the relationships with extrapolation to the entire population, basic statistical tools for describing and making decisions were used. The distribution of the analyzed indicators differed from the normal one (Shapiro–Wilk test); therefore, non-parametric statistics methods were used. The results are presented as: Me = median, and Q_25_–Q_75_ = interquartile range (IQR). In parallel, the usual indicators were used: the arithmetic mean ($\bar{X}$) and standard deviation (±SD). Then, a series of two-way ANOVA tests were performed in order to examine faculties and gender differences in physical activity, satisfaction with life, perceived stress, and coping styles. The significance of the differences in quantitative indicators was assessed by the Mann-Whitney U-test. The three groups were compared using the Kruskal–Wallis test. Pearson’s χ^2^ was used to assess the relationship between the qualitative indicators. Correlations between 0.2 and 0.4 were considered clear, those between 0.4 and 0.6 were considered important, and those between 0.6 and 0.8 were considered considerable. When a “-” appears in the result, it means that the relationship was inversely proportional. The interval estimate of the statistical parameters was determined using the 95% confidence interval. A significance level of *p* < 0.05 was assumed in all of the analyzed cases.

## 3. Results

### 3.1. Characteristics of the Sample

Table 1 presents the characteristics of the sample of respondents according to city and specialization.

### 3.2. Physical Activity

Figure 1 shows the distribution of PA levels in the three groups of students. In terms of the intensity of physical activity, the majority of respondents met the IPAQ criteria [31] for a sufficient and high level. PA, which, in accordance with the recommendations, was recognized as intensive, was stated by 55.2% of the students. In 75.5% of cases, these were student-athletes. High and sufficient PA was determined in 41.1% of the respondents, and insufficient PA was determined in 3.7% of cases (5.8% of pedagogical students and 3.6% of future doctors). Only half of the cases of the students from these two groups fell into the highly active group in terms of PA. The group of student-athletes made up a significant share of the high level of PA.

The total PA index was 4530.7 ± 1769.0 MET-min/week (Me = 2970.0, IQR = 4334). Among the medical students, it turned out to be minimal, at 3427.1 ± 3124.1 (Me = 2464.0, IQR = 2612); among students of pedagogical faculties it was average, 4470.7 ± 4568.8 (Me = 2719.1, IQR = 4881); and among student-athletes it was high, 6271.2 ± 5233.6 MET-min/week (Me = 4729.2, IQR = 5968).

Women prevailed among the respondents (70.4%). The age of the respondents was from 18 to 35 years old ($\bar{X}$ = 19.3; SD = 2.21), with a significant bias towards younger participants. Because the ratio of men to women ranged from 1:1.2 to 1:5, gender was considered in the investigation of MET-min/week index. Men were characterized by a higher level of PA. However, there was also some discrepancy in the identified trend: women representing pedagogy had higher values of MET-min/week for intense physical activity than their male fellow students (Me = 640 MET-min/week for women vs. 480 MET-min/week for men). Statistically significant gender differences were recorded in intense and moderate PA, and walking (Table 2).

The results of the analysis of the weekly time spent on PA indicate that daily intense physical activity—which is accompanied by rapid breathing, an accelerated heartbeat, and sweating—was not often declared by the respondents (only in 14.8% of cases). Moderate PA, which requires medium effort, accompanied by some increase in breathing and without excessive sweating (for example, carrying light objects, cycling at a normal pace, or practicing in amateur sports sections), was stated by 91.0% of the respondents. The third type of analyzed activity was walking during work, at home, during recreation, or as exercise. This activity was noted by almost all of the respondents (99.3%).

The total levels of PA depending on the profile of education, the specialty received, and gender—classified as low, sufficient, advanced and high—are presented as percentages in Table 3.

### 3.3. Life Satisfaction Scale (SWLS)

With regard to the degree of satisfaction with life and considering future specialty, student-athletes were most satisfied with life, at 26.2 ± 5.82 (Me = 27.0, IQR = 7). This indicator turned out to be 1.6 times higher than that of future doctors, at 16.2 ± 6.95 (Me = 16.0, IQR = 9). Pedagogical students were between these two groups, at 24.5 ± 6.65 (Me = 25.0, IQR = 12). Extreme dissatisfaction with life (5–9 points) was noted in 6.0% of cases (among medical students, this indicator was the highest, at 13.8%, while among student-athletes it did not exceed 1%). A high degree of dissatisfaction (10–14 points) was observed in 12.9% of cases, while an average degree of dissatisfaction (15–19 points) was observed in 17.1% of cases. An indifferent answer (20 points), indicating that the respondent does not experience either dissatisfaction with life or satisfaction, was reported among 6.1%. Moderate (21–25 points) and high (26–30) satisfaction were noted in 21.7% and 21.5% of the observations (among student-athletes, this indicator is the highest, at 26.9% and 57.2%, respectively) (Table 4).

The average index of life satisfaction for all of the students surveyed was 22.0 ± 7.84 (Me = 23, IQR = 12). On a three-stage scale, a low degree of satisfaction with life (5–19 points) was noted in a third of observations (36.1%). The results indicating an average score (20–25 points) turned out to be typical for 27.8% of the respondents. The maximum average degree of satisfaction with life was declared in 36.2% of cases, and among student-athletes it reached the level of 57.2%. A slightly lower value of this indicator was found among representatives of pedagogical specialties (46.5%). Among medical students, life satisfaction was experienced by no more than 10% of the respondents.

### 3.4. Perceived Stress Scale (PSS-10)

The subjective perception of the general level of tension in a stressful situation made it possible to evaluate and take into account the efforts made to counteract stress. Table 5 shows summary data characterizing the differences between the students of the three groups according to the results of the Perceived Stress Scale (PSS-10). It can be assumed that the perceived stress of students is mainly associated with overload. Differences are expressed depending on belonging to the study group and gender. There were no significant differences in the “Stress response” subscale.

The relationship between the PSS-10 index and the characteristics of PA and satisfaction with life in the studied groups had certain features. Correlation analysis confirmed a direct correlation (*r* = 0.135, *p* < 0.05) between the level of PA and life satisfaction. Students declaring a high level of PA were characterized by pronounced life satisfaction and high stress resistance. An inverse correlation was also established between the level of PA and the severity of perceived stress (*r* = −0.122, *p* < 0.05), indicating a decrease in the level of stress with an increase in PA. With an increase in the perception of stress, life satisfaction decreases, which is confirmed by an even more significant negative relationship between these indicators (*r* = −0.275, *p* < 0.05). For a more detailed clarification of scenarios for coping with stress, it was necessary to prioritize the chosen methods, which was achieved by evaluating coping strategies.

### 3.5. The Choice of Coping Strategies

Table 6 shows the rating distribution of the frequency of choice of the coping strategies among the students surveyed, as well as significant differences in the groups.

Figure 2 shows the distribution of the frequency of choice of coping strategies by students, taking into account belonging to one of the three groups according to the profile of education.

All 14 coping strategies were grouped into three integral ones. Thus, the active strategy included active coping, planning, and positive reframing. The emotion-oriented support-seeking strategy included religion, the use of emotional and instrumental support, the inability to contain emotions, venting them, and a strategy of self-condemnation. The avoidance scenario consisted of acceptance or denial of the situation, self-distraction with other activities, behavioral withdrawal, and a tendency to use alcohol and psychoactive substances.

When choosing the three integral coping strategies, it was noted that students almost equally, and more often, resort to active coping (Complex coping scenario “Active coping” (scales 1–3)). Among the rarely used coping strategies were avoidance, blaming oneself, denial of the reality of a stressful situation, ignoring it, turning to religion, and behavioral withdrawal (“helplessness”, “submission”, “refusal of efforts”, and Complex coping scenario “Avoidance coping” (scales 4, 5, 9, 10, 12, 13). The respondents associated coping with stress with turning to alcohol and other psychoactive substances. Reliably often, future doctors choose coping strategies aimed at overcoming the current situation by seeking support or emotionally-oriented coping (Complex coping scenario “support-seeking/emotion-oriented coping” (scales 6–8, 11, 14)). In student-athletes, active overcoming and minimal indicators of the other two coping strategies prevail.

Correlation analysis made it possible to establish the relationship between PA, life satisfaction, the perception of stress, and the choice of complex coping strategies. Students declaring a high level of PA were characterized by high stress resistance and the choice of active coping strategies; they significantly less often resorted to a support search strategy focused on emotions, the impossibility of restraining them, self-condemnation, or avoidance strategies (Table 7).

By combining the most important indicators of the correlation analysis (physical activity level, SWLS, PSS-10 and priority coping strategies), a classification of behaviors aimed at coping with a stressful situation was proposed. The classification provides the possibility of the individual assignment of the respondent to a group with optimal, acceptable, satisfactory and risky behavior (Figure 3). The optimal behavior model was a declaration of high and sufficient PA (IPAQ), expressed life satisfaction (SWLS), and was characterized by a low level of perceived stress (PSS).

The model of acceptable behavior was characterized by the sufficient PA of the respondents, an average degree of life satisfaction, and an average level of perceived stress. Satisfactory behavior occupied a niche between acceptable and risky behaviour. Risky behaviour was characterized by insufficient PA, a low level of life satisfaction, and a tendency to a high level of perceived stress (Table 8).

The frequency of occurrence of optimal and acceptable behavior significantly differed between the surveyed groups according to their future profession. Gender differences were found only in student-athletes (Table 9).

## 4. Discussion

In order to successfully adapt to the new conditions that have arisen as a result of extraordinary psychotraumatic events (the COVID-19 pandemic), a person needs to endure and survive this situation, take into account external circumstances, and make lifestyle changes [41,42]. The effectiveness of adaptation is provided by resources that are contained in both personal and external social factors. In this study, for the first time in a comparative aspect, complex indicators of adaptation in the context of the COVID-19 pandemic were compared between Belarusian students of various educational profiles. Particular attention was paid to the indicators of PA, life satisfaction, the significance of perceived stress, and coping strategies for dealing with stress. In Belarus, unlike most countries of the European Union, strict restrictive measures have been abandoned. Belarusian students were expected to experience lower levels of physical activity, dissatisfaction with life, and higher levels of stress and anxiety than before the pandemic. On the other hand, it was assumed that the severity of the consequences of anti-epidemic measures would not have significantly affected the mental health of student youth.

Because the ratio of men to women ranged from 1:1.2 to 1:5, gender was considered in the investigation of PA. Therefore, differences in PA levels were also interpreted by gender. Men were characterized by a higher level of PA. However, there was also some discrepancy in the identified trend, in which women representing pedagogy had higher values of intense PA than their male fellow students. Significant differences in the studied indicators of adaptation depending on the profile of training have been established. During the pandemic, the majority of students noted a decrease in their level of PA; however, this affected student-athletes to a lesser extent, who—to the extent possible—continued playing sports, and demonstrated an increased (13.9%) and high level of PA (75.5%). In 5% of cases, pedagogical students and future doctors declared a low level of PA, which was not typical of student athletes. According to Hall et al., there are two pandemics occurring simultaneously: the coronavirus pandemic and the physical inactivity pandemic [43]. Zaworski et al., in their study of 688 people from Poland, reported a significant reduction in the frequency and duration of PA [44]. In this regard, a Canadian study provided evidence for the adverse effects of the COVID-19 outbreak among young people, demonstrating that only 2.6% practiced 60 min of moderate-to-vigorous PA per day [45]. In our studies, compliance with the recommended WHO [4] indicators of daily moderate and intense PA was noted in 17.0% of cases. Forced physical inactivity was of particular concern in the context of the success of the educational process, because it is known that regular physical exercise among university students reduces the incidence of depression and increases life satisfaction [46], which contributes to the successful acquisition and assimilation of knowledge. PA is a protective factor: it helps to maintain life satisfaction and reduces the degree of perception of stress [47]. This is consistent with the data on PA modulating the levels of hormones, amino acids, and neurotransmitters, which reduces psychological stress. Regular PA of adequate intensity is suggested as an aid to strengthening the immune system to COVID-19 [48,49].

As in other studies, our results indicate a positive effect of elevated and high levels of PA on wellbeing during a pandemic. Maintaining the PA regime in this situation may require special willpower and motivational skills in order to overcome the barriers that are characteristic of the pandemic. Among the recommendations, an important place is occupied by the promotion of PA even at the level of recreational activity, which minimizes exposure to stress and allows you to maintain physical fitness, body performance, and an optimal bodyweight [50].

Despite the minimization of restrictions in Belarus [51], the stratification in terms of the level of PA between students of various specialties is expressed significantly. Due to the specifics of presenting information to the population about the epidemiological situation, pedagogical students and the majority of student-athletes refused to believe in what had happened, denied the reality, and rarely radically changed their lifestyle. The important role of the mass media (media), and the opinion of the medical community and health authorities, which directly or indirectly influenced the behavior of the population and modeled the situation through the provision of information, came into play. Due to the subjectivity in the presentation of information, it is recommended to use the media and social networks with caution in the situation of COVID-19 [52].

The level of life satisfaction among medical students was lower than that in other groups of students, which can be associated with the influence of the media and there being some discrepancy between information on morbidity and information obtained directly during practical classes in healthcare institutions. Among student-athletes, this score was the highest. It turned out to be 1.6 times higher than that of future doctors. The pedagogical students were between student-athletes and medical students. It should be noted that the respondents whose PA was characterized as high made up more than half (53.0%) of those were most satisfied with life. In half of the cases in which the students declared an insufficient level of PA and a passive lifestyle, they noted a low degree of satisfaction with life (46.9%). In the group of physically active respondents, a quarter of students (24.6%) did not receive satisfaction from life.

It is assumed that the decrease in PA and dissatisfaction with life are due to restrictions in physical exercises based on sports clubs and fitness centers. Home workouts could not replace classes at the sports center [53] but still leveled the deficit of movement [54]. Perhaps this is why student-athletes were characterized by a higher degree of life satisfaction and less pronounced indicators of perceived stress. Many authors note that tough anti-epidemic measures have reduced the level of training of athletes, life satisfaction, and mental health in general compared with the stages of life normalization [55]. In our study, there was no significant decrease in life satisfaction and a pronounced deterioration in sports performance in this group of students. In turn, the group of future doctors is distinguished by a pronounced degree of the perception of stress, which is consistent with the studies of other authors [56,57], and suggests that the specifics of training can potentiate a high level of stress in both medical and pedagogical students [58]. Students of medical specialties had a significantly higher perception of stress than their peers from the other groups. The minimum indicator was noted among student-athletes. The severity of the indicator in female students prevails over the data of male students. The results of the assessment on the “Overload subscale” confirmed a higher level of stress among medical students. The indicators of the female students were also significantly higher than those of the male students. Because stress causes psychological responses, it is important to have an idea about the options for responding to an extreme situation. The study of human behavior in stressful situations makes it possible to identify coping mechanisms that determine the success and failure of adaptation. The appeal of certain strategies for coping with stress in a pandemic situation among students is associated with several factors: the specifics of the conditions and anti-epidemiological measures, the characteristics of the training profile, and gender.

According to our data, the majority of Belarusian students, in order to cope with the stress caused by the pandemic, chose coping strategies of active coping and problem-oriented methods aimed at changing the situation, often combined with elements of emotionally-oriented coping. Coping methods of avoidance and withdrawal were used much less frequently. Behavioral disengagement (“helplessness”, “submission”, “refusal of efforts”), conversion to religion, and the use of alcohol and other psychoactive substances were even rarer. Student-athletes differed from the other groups of respondents in their pronounced active position in overcoming stress.

Our results show general trends in the choice of coping strategies for students of various specialties: the low use of “sedatives” and alcohol consumption, the rare use of religion, and the more frequent use of active coping and planning, which is consistent with the results of other studies [59]. However, there are differences in the frequency of choosing these strategies. The uncertainty of the situation with restrictions and the often direct contact with the “problem” among medical students reduced the proportion of requests for planning and developing a strategy for active action. Medical university students, especially those who showed the worst results in terms of PA and life satisfaction, often used emotionally-oriented coping strategies associated with outbursts of negative emotions—such as anger, irritation, frustration or sadness—in order to cope with stress. This result is consistent with studies by other authors [60], confirming that medical students are usually more susceptible to stress [61]. Almost half of all medical students may experience burnout while studying at a university [62], and this may be underestimated when studying the impact of stress on the bodies of future doctors [63]. An additional stress load can be associated with exposure to other factors: the intensity of theoretical and practical training, a significant amount of knowledge gained, a diverse combination of academic deadlines, and the experience of increased responsibility for learning [64]. A Russian study reported that 69.1% and 4.9% of medical students were at medium and high risk of adverse effects of stress on the body [56]. The revealed trend in the choice of coping strategies by students of various specialties can characterize, on the one hand, the characteristics of students and their readiness to work in critical, extreme situations, and on the other hand, the specifics of education and professional training. Students of medical specialties more often note an appeal to problem-oriented coping, and representatives of pedagogical and humanitarian areas show more pronounced coping by avoiding problems, which was also noted by other authors [65,66].

Belonging to the group of student-athletes is associated with the maximum life satisfaction, a high level of PA, a minimal perception of stress, and a rare use of the integral coping strategies “Avoidance coping” and “Support-seeking/emotion-oriented coping”. Medical students are characterized by a high frequency of the perception of stress and a more frequent use of the integral coping strategies “Avoidance coping” and “Support-seeking/emotion-oriented coping”.

The results obtained allow us to identify trends in the use of coping strategies depending on gender. Female students more often turn to coping strategies associated with emotions and concentration on negative experiences. They are also characterized by mental and behavioral withdrawal, and the desire for social and instrumental support. They want to receive advice, help, and information, which may be due to the uncertainty of the situation itself; they have a higher level of perception of stress compared to students, as recorded in our studies and the works of other authors [2]. Male students (athletes) are more likely to take active actions [67], which was also recorded by us. Moreover, the male respondents showed a low level of concentration on emotions and a frequent use of humor. There were no significant differences in the choice of coping strategies among men in the three groups of respondents. Women who were involved in sports significantly—more often than their peers from other groups—used active coping strategies.

Our attempt to link PA options, life satisfaction, the perception of stress, and measures to overcome it together can help in individual diagnostics of mental health and, at the screening stage, can identify a risk group based on behavior in a stressful situation, which is combined with recommendations to give priority to a set of indicators of physical and mental health [68].

## 5. Conclusions

The majority of the students studying at universities in Belarus demonstrated a high and elevated level of PA. Student athletes had the highest level of PA. Representatives of medical specialties had a deficit of PA in comparison with the students of pedagogical faculties and students of faculties of physical culture. Among the medical and pedagogical specialties, 5% of the respondents were classified as not being physically active enough. Based on these results, possible intervention programs and tailored health promotion strategies can be generated. There is a strong need to promote being physically active and inculcating the habit of spending free time actively, especially among students of medical and pedagogical faculties.

The maximum satisfaction from life was noted among student-athletes. The correlation confirming the increase in the level of PA with the increase in life satisfaction was pronounced.

The future doctors from the three groups of students surveyed were characterized by the highest level of perceived stress. In contrast, the lowest score was for student-athletes. Most of the respondents used coping strategies of active coping, planning, seeking emotional support, and positive reassessment and development. The choice of a strategy of avoidance, the avoidance of the problem, and the search for support or emotionally-oriented overcoming was practically not met among the student-athletes, but it was noted among future doctors. Pedagogical students, regarding the choice of coping strategies, occupied an intermediate position between future doctors and student athletes.

Depending on the severity of the declared PA, life satisfaction, the perception of stress, and the priority choice of coping strategies, all of the respondents can be divided into four groups that differ in terms of their behavior aimed at overcoming stress (optimal, acceptable, satisfactory and risky). Taking into account this classification, the most favorable opportunities for coping with stress were found among student-athletes and, in descending order of effect, among representatives of the pedagogical and medical students.

## 6. Recommendations

In order to adequately overcome stress in the dynamically changing situation of COVID-19, it is necessary to monitor mental and physical health, develop a common strategy for involving student youth (especially future medics) in active physical education, increase the time allotted for organized and independent physical activity, and promote a healthy lifestyle.

The situation could be improved if the medical students were better trained to improve their coping strategies and deal with the specifically stressful aspects (there are specific programmes designed to enhance future physician’ self-efficacy). The results provide significant information for both healthcare organizations and educational establishments, as they can be used as grounds for suggesting activities aimed at maintaining students’ wellbeing and providing wider opportunities for young people to pursue a healthy lifestyle.

### Limitations of the Study

Despite the scope, representativeness, and versatility of the study, there are some limitations on the interpretation of the results. The data obtained from the three groups of students of various specialties can be conditionally generalized and extrapolated to the entire population of students of Belarusian universities. Due to the subjective approach, the choice of answers may not always correctly characterize the respondent or contribute to a correct assessment of the situation. It is possible that some students expressed their negative emotions through the questionnaire, and that stress was overestimated due to fear associated with a lack of information about the current situation and the consequences of COVID-19. On the other hand, psychiatric pathology was not controlled for in this study, such that the average level of stress perception may be increased by the participation of the respondents with a past individual history of stress disorders brought on from the first waves of the pandemic, as well as in connection with illness, self-isolation and quarantine. We are currently finalizing the results of a unified cross-border adaptation study of more than 5000 students from closely spaced cities in Poland, Russia, Lithuania and Belarus, which will allow us to conduct a comparative analysis of physical activity, life satisfaction and coping strategies during the COVID-19 pandemic among medical students and non-medical universities in cities on both sides of the eastern border of the European Union, i.e., in Poland, Lithuania, Russia, Belarus and Ukraine, in the context of dynamic changes in the situation.

## Figures and Tables

**Figure 1 ijerph-19-08629-f001:**
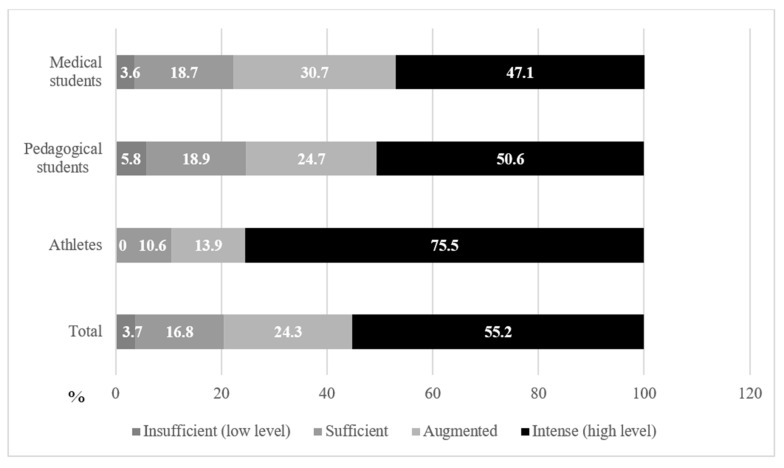
Proportion of different types of physical activity (PA) declared by the respondents.

**Figure 2 ijerph-19-08629-f002:**
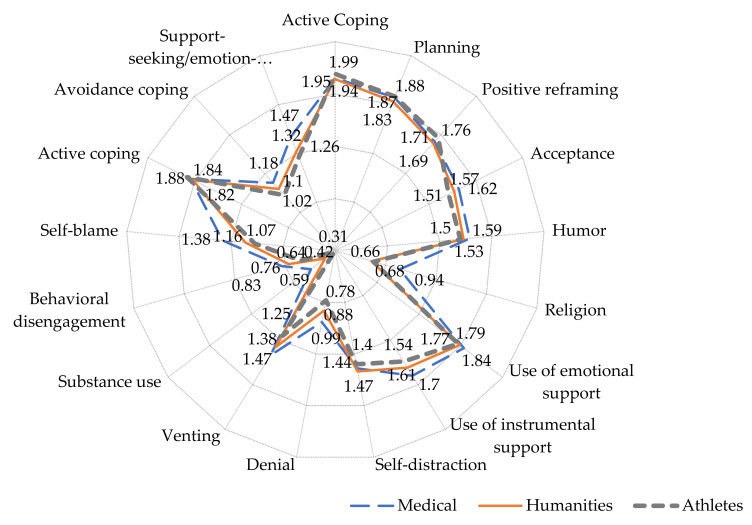
Manifestations of coping strategies among the students in the three groups.

**Figure 3 ijerph-19-08629-f003:**
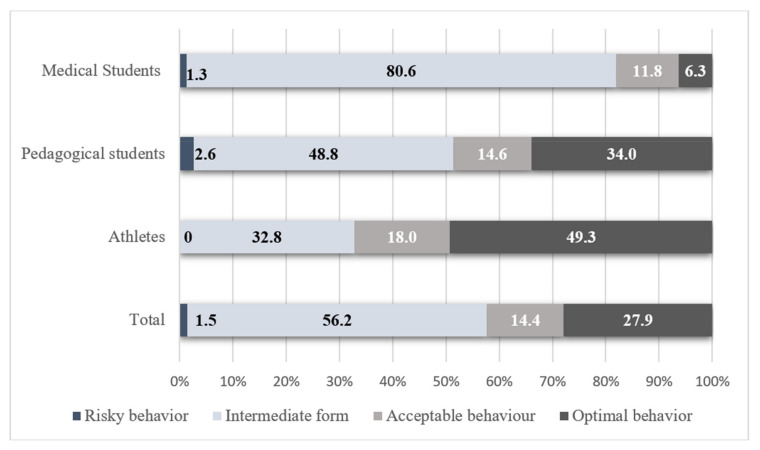
Share of behavior patterns depending on the group of the respondents.

**Table 1 ijerph-19-08629-t001:** Characteristics of the study population.

	Universities in Cities *N* (%)								Total Sample *N* = 1769
	Grodno, *N* = 852 (48.1%)			Minsk, *N* = 375 (21.2%)	Brest, *N* = 205 (11.6%)	Vitebsk, *N* = 337 (19.1%)			
	Specializations That Students Receive at the University (Faculties)								
	Medical (*N* = 514)	Pedagogical (*N* = 211)	Athletes (*N* = 127)	Pedagogical (*N* = 375)	Athletes (*N* = 205)	Medical (*N* = 103)	Pedagogical (*N* = 151)	Athletes (*N* = 83)	
**Male (*n*, %, 95%CI)**	126, 24.5 (20.8–28.2)	55, 26.1 (20.2–32.0)	58, 45.7 (37.0–54.3)	79, 21.1 (16.9–25.2)	108, 52.7 (45.9–59.5)	26, 25.2 (16.9–33.6)	29, 19.2 (12.9–25.5)	42, 50.6 (39.8–61.4)	523, 29.6 (27.4–31.7)
**Female (*n*, %, 95%CI)**	388, 75.6 (71.9–79.4)	156, 73.9 (68.0–79.9)	69, 54.3 (45.7–63.0)	296, 78.9 (74.8–83.1)	97, 47.3 (40.5–54.2)	77, 74.8 (66.4–83.2)	122, 80.8 (74.5–87.1)	41, 49.4 (38.6–60.2)	1246, 70.4 (68.3–72.6)
**Age, mean (years ± SD)**	18.9 ± 2.11	19.5 ± 3.01	20.8 ± 1.72	18.9 ± 1.21	19.9 ± 2.28	19.3 ± 1.44	20.1 ± 3.37	19.3 ± 1.7	19.3 ± 2.21
**Quarantine or strict self- isolation (*n*, %, 95%CI)**	174, 33.9 (29.6–37.8)	79, 37.4 (30.9–44.0)	49, 38.3 (29.9–46.7)	179, 47.7 (42.7–52.8)	63, 30.7 (24.4–37.1)	39, 37.9 (28.5–47.2)	53, 35.1 (27.5–42.7)	27, 32.5 (22.5–42.6)	662, 37.4 (31.8–36.2)
	302, 35.5 (32.1–38.5)					119, 35.3 (30.2–40.4)			
**Vaccinated against** **COVID- 19 (*n*, %, 95%CI)**	280, 54.5 (50.1–58.7)	60, 28.4% (22.4–34.5)	57, 44.5 (35.9–53.1)	93, 24.8 (20.4–29.2)	122, 59.5 (52.8–66.2)	68, 66.0 (56.9–75.2)	78, 51.7 (43.7–59.6)	36, 43.4 (32.7–54.0)	794, 44.8 (42.5–47.2)
	397, 46.6 (43.1–49.8)					182, 54.0 (48.7–59.3)			
**Contact with persons who has been diagnosed COVID-19**	174, 33.9 (29.8–37.9)	65, 30.8 (24.6–37.0)	30, 23.6 (16.2–31.0)	132, 35.2 (30.4–40.0)	56, 27.3 (21.2–33.4)	49, 47.6 (37.9–57.2)	48, 31.8 (24.4–39.2)	25, 30.1 (23.3–40.0)	579, 32.7 (30.5–34.9)
	269, 31.6 (28.5–34.7)					122, 36.2 (31.1–40.3)			
**Diagnosed with COVID 19 (infection with SARS-CoV-2) (*n*, %, 95%CI)**	159, 31.4 (26.9–34.9)	60, 28.4 (22.4–34.5)	26, 20.5 (13.5–27.5)	115, 30.7 (26.0–35.3)	47, 22.9 (17.2–28.7)	39, 37.9 (28.5–47.2)	40, 26.5 (19.5–33.5)	18, 21.7 (12.8–30.6)	504, 28.6 (26.4–30.6)
	245, 29.0 (26.0–32.1)					97, 28.8 (23.9–33.6)			
**Declared regular physical activity (*n*, %, 95%CI)**	134, 26.1 (22.3–29.9)	63, 30.3 (24.1–36.5)	79, 62.5 (54.1–70.9)	64, 16.8 (13.0–20.6)	122, 59.5 (52.8–66.2)	37, 35.9 (26.7–45.2)	39, 25.8 (18.9–32.8)	60, 72.3 (62.7–81.9)	599, 33.9 (31.7–36.1)
	276, 32.4 (29.3–35.5)					136, 40.4 (35.1–45.6)			

Note: *N* is the number of observations; % is the percentage of the total number of study participants in a given group; 95%CI, 95-percent confidence interval; SD, standard deviation.

**Table 2 ijerph-19-08629-t002:** Descriptive statistics of the main types of PA of Belarusian students.

Physical Activity	Group	Gender	$\bar{X}$ ± SD	Me, IQR	*p*-Test Probability Value Calculated Using Mann-Whitney Test and Kruskal-Wallis Test	
**Intensive**	Medical Students [1]	Male [4]	1465.8 ± 1565.0	1080; 1520	*p*_[1, 2]_ < 0.001 *p*_[1, 2, 3]_ < 0.001	*p*_[4, 5]_ < 0.01
		Female [5]	1144.1 ± 1803.6	960; 1120		
	Pedagogical students [2]	Male [4]	1058.9 ± 2371.6	480; 1000		*p*_[4, 5]_ < 0.05
		Female [5]	1219.9 ± 2074.0	640; 1200		
	Athletes [3]	Male [4]	2895 ± 3005	2240; 3280		*p*_[4, 5]_ < 0.01
		Female [5]	2341.6 ± 2780	1440; 2600		
**Moderate**	Medical Students [1]	Male [4]	694.6 ± 1052.3	480; 560	*p*_[1, 2]_ < 0.01 *p*_[1, 2, 3]_ < 0.001	*p*_[4, 5]_ < 0.01
		Female [5]	519.6 ± 738.2	320; 440		
	Pedagogical students [2]	Male [4]	736.8 ± 1734	308; 640		*p*_[4, 5]_ > 0.1
		Female [5]	613.4 ± 1167	320; 480		
	Athletes [3]	Male [4]	1361.6 ± 1816	840; 1080		*p*_[4, 5]_ > 0.1
		Female [5]	1125.8 ± 1579	720; 1160		
**Walking**	Medical Students [1]	Male [4]	1546.0 ± 1966.1	840; 980	*p*_[1, 2]_ < 0.001 *p*_[1, 2, 3]_ < 0.001	*p*_[4, 5]_ > 0.1
		Female [5]	1488.2 ± 1705.3	840; 1230		
	Pedagogical students [2]	Male [4]	2546 ± 2862	1260; 3240		*p*_[4, 5]_ > 0.1
		Female [5]	2387 ± 2797	1260; 1920		
	Athletes [3]	Male [4]	2105 ± 2665	1200; 660		*p*_[4, 5]_ > 0.1
		Female [5]	2380.3 ± 2645	1260; 660		
**Total physical activity**	Medical Students [1]	Male [4]	3706.4 ± 3039.0	2790; 3184	*p**_[_*_1, 2]_ < 0.001 *p*_[1, 3]_ < 0.001 *p**_[_*_2, 3]_ < 0.001 *p*_[1, 2, 3]_ < 0.001	*p*_[4, 5]_ < 0.01
		Female [5]	3141.4 ± 3092.0	2275; 2400		
	Pedagogical students [2]	Male [4]	4341.6 ± 4415.8	2850; 5440		*p*_[4, 5]_ > 0.1
		Female [5]	4220.0 ± 4435.0	2570; 4380		
	Athletes [3]	Male [4]	6361.5 ± 5585.8	4680; 5540		*p*_[4, 5]_ > 0.1
		Female [5]	5747.8 ± 4490.2	4383; 5990		

**Table 3 ijerph-19-08629-t003:** Declared level of physical activity by future profession and gender.

Level of Declared Physical Activity	Groups of Students							
	Medical (*N* = 617)		Pedagogical (*N* = 737)		Athletes (*N* = 416)		Total (*N* = 1769)	
	Male (*n* = 152)	Female (*n* = 465)	Male (*n* = 163)	Female (*n* = 574)	Male (*n* = 209)	Female (*n* = 207)	Male (*n* = 523)	Female (*n* = 1246)
**Insufficient** **(*n*, %, 95%CI)**	5, 3.3 (0.1–5.2)	18, 3.9 (2.1–5.6)	13, 8.0 (3.8–12.1)	30, 5.2 (3.4–7.1)	0	0	18, 3.3 (1.7–4.8)	48, 3.9 (2.8–4.9)
	23 3.7 (2.1–5.0)		43 5.8 (4.1–7.5)		0		66 3.7 (2.8–4.6)	
**Sufficient** **(*n*, %, 95%CI)**	25, 16.6 (10.6–22.5)	90, 19.4 (15.8–22.9)	32 19.6 (13.5–25.7)	107, 18.6 (15.5–21.8)	21, 10.1 (6.0–14.1)	23, 11.1 (6.8–15.4)	78, 14.9 (11.9–18.0)	220, 17.7 (15.5–19.8)
	115 18.7 (15.6–21.8)		407 55.2 (51.6–58.8)		44 10.6 (7.6–13.5)		298 16.8 (15.1–18.6))	
**Augmented** **(*n*, %, 95%CI)**	35, 23.2 (16.5–29.9)	154, 33.1 (28.8–37.4)	32, 19.6 (13.5–25.7)	150, 26.1 (22.5–29.7)	25, 12.0 (7.6–16.7)	33, 15.9 (11.0–20.9)	92, 17.6 (14.3–20.9)	337, 27.0 (24.6–29.5)
	189 30.7 (27.0–34.3)		182 24.7 (21.6–27.8)		58 13.9 (10.6–17.3)		429 24.3 (22.3–26.3)	
**High** **(*n*, %, 95%CI)**	87, 48.3 (49.7–65.5)	203, 47.1 (39.2–48.2)	86, 52.8 (45.1–60.4)	287, 50.0 (45.9–54.1)	163, 78.0 (72.4–83.6)	151, 72.9 (66.9–79.0)	336, 64.2 (60.1–68.4)	641, 51.4 (48.7–54.2)
	290 39.6 (43.1–51.0)		373 50.6 (47.0–54.2)		314 75.0 (71.4–79.6)		977 55.2 (52.9–57.6)	

**Table 4 ijerph-19-08629-t004:** SWLS scores for the three groups of students surveyed (*n*, %, 95%CI).

Groups of Students	Gender	SWLS (Points)		Extremely Dissatisfied	Dissatisfied	Slightly Dissatisfied	Neutral	Slightly Satisfied	Satisfied	Extremely Satisfied
				Scores						
		M ± SD	Me; IQR	5–9	10–14	15–19	20	21–25	26–30	31–35
**Medical Students (*N* = 617)** [1]	Male (*n* = 152)	16.0 ± 6.99	15.0; 9	23 15.2 (9.5–21.0)	43 28.5 (21.3–35.7)	40 25.8 (18.9–32.8)	12 7.9 (3.6–12.3)	22 14.6 (8.9–20.2)	5 3.3 (0.5–6.2)	7 4.6 (1.3–8.0)
	Female (*n* = 465)	16.3 ± 6.94	16.0; 9	62 13.3 (10.2–16.4)	140 30.1 (25.9–34.3)	117 25.2 (21.2–29.1)	35 7.5 (5.1–9.9)	64 13.8 (10.6–16.9)	25 5.4 (3.3–7.4)	22 4.7 (2.8–6.7)
	Total	16.2 ± 6.95	16.0; 9	85 13.8 (11.1–16.5)	183 29.7 (26.1–33.3)	157 25.3 (21.9–28.8)	47 7.6 (5.5–9.7))	86 14.0 (11.2–16.7)	30 4.9 (3.2–6.6)	29 4.7 (3.0–6.4)
**Pedagogical students (*N* = 737)** [2]	Male (*n* = 163)	24.7 ± 7.0	25.0; 10	4 2.5 (0.1–4.8)	9 5.5 (2.0–9.0)	26 16.0 (10.3–21.6)	12 7.4 (3.4–11.4)	36 22.1 (15.7–28.5)	41 25.2 (18.5–31.8)	35 21.5 (15.2–27.8)
	Female (*n* = 574)	24.4 ± 6.55	25.0; 9	15 2.6 (1.3–3.9)	24 4.2 (2.5–5.8)	83 14.5 (11.6–17.3)	36 6.3 (4.3–8.3)	149 26.0 (22.4–29.6)	168 29.3 (25.6–33.0)	99 17.2 (14.2–20.3)
	Total	24.5 ± 6.65	25.0; 10	19 2.6 (1.4–3.7)	33 4.5 (3.0–6.0)	109 14.8 (12.2–17.4)	48 6.5 (4.7–8.3)	185 25.1 (22.0–28.2)	209 28.4 (25.1–31.6)	134 18.2 (15.4–21.0)
**Athletes** **(*N* = 415)** [3]	Male (*n* = 208)	26.7 ± 5.94	27.0; 8	3 1.4 (0–3.1)	4 1.9 (0.1–3.8)	17 8.1 (4.4–11.8)	5 2.4 (0.3–4.5)	56 26.8 (20.8–32.8)	67 32.1 (25.7–38.4)	56 27.3 (21.2–33.3)
	Female (*n* = 207)	25.7 ± 5.66	26.0; 8	0	8 3.9 (1.2–6.5)	21 10.1 (6.0–14.3)	8 3.9 (1.2–6.5)	56 27.1 (21.0–33.1)	75 36.2 (29.7–42.8)	39 18.8 (13.5–24.2)
	Total	26.2 ± 5.82	27.0; 7	3 0.7 (0–1.5)	12 2.9 (1.3–4.5)	38 9.1 (6.4–11.9)	13 3.1 (1.5–4.8)	112 26.9 (22.7–31.2)	142 34.1 (29.6–38.7)	95 23.1 (19.0–27.1)
**Total** **(*N* = 1769)**	Male (*n* = 523)	23.0 ± 7.99	24.0; 13	30 5.7 (3.8–7.7)	56 10.7 (8.1–13.4)	83 15.7 (12.6–18.8)	29 5.5 (3.6–7.5)	114 21.1 (18.3–25.3)	113 21.6 (18.1–25.1)	98 18.9 (15.6–22.3)
	Female (*n* = 1246)	21.6 ± 7.75 *	22.0; 12	77 6.2 (4.8–7.5)	172 13.8 (11.9–15.7)	221 17.7 (15.6–19.9)	79 6.3 (5.0–7.7)	269 21.6 (19.3–23.9)	268 21.5 (19.2–23.8)	160 12.8 (11.0–14.7)
	Total	22.0 ± 7.84	23.0; 12	107 6.0 (4.9–7.2)	228 12.9 (11.3–14.5)	304 17.1 (15.4–18.9)	108 6.1 (5.0–7.2)	383 21.7 (19.7–23.6)	381 21.5 (19.6–23.5)	258 14.6 (13.0–16.3)
	*p*_[1, 2]_ < 0.001 *; *p*_[1, 3]_ < 0.001 *; *p*_[2, 3]_ < 0.001 *; *p*_[1, 2, 3]_ < 0.001 **									

* Mann–Whitney. ** Kruskal–Wallis.

**Table 5 ijerph-19-08629-t005:** Summary of the data describing the differences in the scores obtained by students of both genders in the three groups on the Perceived Stress Scale and its subscales.

Theorem Scale	Group	Gender	Mean Score	Me, IQR	*p*-Test Probability Value Calculated Using Mann-Whitney Test and Kruskal-Wallis Test
**Perceived stress**	Medical students [1]	Male	18.9 ± 7.66	20; 7	*p**_[_*_1, 2]_ < 0.001 (U = 213,981); *p*_[1, 3]_ < 0.001 (U = 98,220); *p*_[2, 3]_ < 0.001 (U = 126,424);
		Female	20.2 ± 7.55	21; 6	
		Total	19.9 ± 7.59	20; 7	
	Pedagogical students [2]	Male	17.2 ± 7.45	19; 8	
		Female	19.8 ± 7.50	20; 7	
		Total	19.2 ± 7.56	20; 8	
	Athletes [3]	Male	16.1 ± 6.96	18; 7.5	
		Female	18.7 ± 7.10	20; 7	
		Total	17.4 ± 7.14	19; 8	
	Total	Male	17.3 ± 7.39	19; 7	Test Kruskala-Wallisa: H = 40.6 *p*_[1, 2, 3]_ < 0.001
		Female	19.8 ± 7.46	20; 7	
		Total	19.0 ± 7.53	20; 7	
**Overload subscale**	Medical students [1]	Male	9.7 ± 5.73	10; 6	*p*_[1, 2]_ < 0.001 (U = 213,981); *p*_[1, 3]_ < 0.001 (U = 91,518); *p*_[2, 3]_ < 0.001 (U = 123,246)
		Female	11.7 ± 5.61	12; 7	
		Total	11.2 ± 5.70	12; 8	
	Pedagogical students [2]	Male	8.8 ± 5.78	9; 7	
		Female	10.9 ± 5.91	11; 7	
		Total	10.4 ± 5.95	11; 8	
	Athletes [3]	Male	7.0 ± 4.95	6; 8	
		Female	10.1 ± 5.46	10; 6	
		Total	8.5 ± 5.43	8; 8	
	Total	Male	8.3 ± 5.56	8; 8	Test Kruskala-Wallisa: H = 59.0 *p*_[1, 2, 3]_ < 0.001
		Female	11.1 ± 5.75	12; 8	
		Total	10.2 ± 5.83	11; 8	
**Stress response**	Medical students [1]	Male	9.2 ± 4.19	10; 4.5	N/S
		Female	8.6 ± 3.66	9; 3	
		Total	8.7 ± 3.80	9; 3	
	Pedagogical students [2]	Male	8.5 ± 4.04	8; 6	
		Female	8.9 ± 3.86	9; 5	
		Total	8.8 ± 3.91	9; 5	
	Athletes [3]	Male	9.1 ± 4.26	10; 5	
		Female	8.7 ± 3.61	9; 5	
		Total	8.9 ± 3.95	9; 5.5	
	Total	Male	9.0 ± 4.17	9; 6	
		Female	8.7 ± 3.75	9; 5	
		Total	8.8 ± 3.88	9; 5	

**Table 6 ijerph-19-08629-t006:** Results of the rating choice of the strategies for coping with perceived stress.

Coping-Strategy, Rating Number (Scale Number)	$\bar{X}$ ± SD, *Me*; IQR	Significant Differences in the Groups *
**1. (1) Active Coping**	1.95 ± 0.76; 2.0; 1.0	not significant
**2. (2) Planning**	1.85 ± 0.78; 2.0; 1.0	not significant
**3. (7) Use of emotional support**	1.80 ± 0.84; 2.0; 1.5	not significant
**4. (3) Positive reframing**	1.71 ± 0.83; 2.0; 1.0	not significant
**5. (8). Use of instrumental support**	1.62 ± 0.80; 1.5; 1.0	*p*_[1, 3]_ < 0.001; *p*_[1, 2, 3]_ < 0.001 **
**6. (4) Acceptance**	1.58 ± 0.78; 1.5; 1.0	*p*_[1, 3]_ < 0.05; *p*_[1, 2, 3]_ < 0.05 **
**7. (5) Humor**	1.54 ± 0.82; 1.5; 1.0	not significant
**8. (9) Self-distraction**	1.44 ± 0.76; 1.0; 0.5	not significant
**9. (11) Venting**	1.38 ± 0.69; 1.5; 1.0	*p*_[1, 3]_ < 0.001; *p*_[2, 3]_ < 0.001; *p*_[1, 2, 3]_ < 0.001 **
**10. (14) Self-blame**	1.21 ± 0.86; 1.0; 1.5	*p*_[1, 2]_ < 0.05; *p*_[1, 3]_ < 0.001; *p*_[1, 2, 3]_ < 0.001 **
**11 (10) Denial**	0.89 ± 0.77; 1.0; 1.5	*p*_[1, 2]_ < 0.001; *p*_[1, 3]_ < 0.001; *p*_[1, 2, 3]_ < 0.01 **
**12 (6) Religion**	0.76 ± 0.88; 0.5; 0.5	*p*_[1, 2]_ < 0.001; *P*_[1, 3]_ < 0.001; *p*_[1, 2, 3]_ < 0.001 **
**13 (13) Behavioral disengagement**	0.75 ± 0.68; 0.5; 1.0	*p*_[1, 2]_ < 0.05; *P*_[1, 3]_ < 0.001; *p*_[2, 3]_ < 0.01; *p*_[1, 2, 3]_ < 0.001 **
**14. (12) Substance use**	0.45 ± 0.72; 0; 1.0	*p*_[1, 2]_ < 0.001; *p*_[1, 3]_ < 0.001; *p*_[2, 3]_ < 0.001; *p*_[1, 2, 3]_ < 0.001 **

Note: * Groups: medical students [1], pedagogical students [2], and athletes [3]. ** *p*-test probability value calculated using the Mann-Whitney test and Kruskal-Wallis test.

**Table 7 ijerph-19-08629-t007:** Pearson’s correlation coefficients between of the main diagnostic indicators.

	Physical Activity Level	SWLS	PPS-10	Active Coping	Avoidance Coping
**SWLS**	0.104				
**PPS-10**	−0.007 *	−0.165			
**Active coping**	0.079	0.069	0.221		
**Avoidance coping**	−0.056	−0.209	0.207	0.180	
**Support-seeking/emotion-oriented coping**	−0.048	−0.134	0.311	0.458	0.629

Note: * The correlation coefficient is unreliable, *p* > 0.05; in other cases, it is significant *p* < 0.05.

**Table 8 ijerph-19-08629-t008:** The classification of behavior models aimed at coping with stress.

Behavioral Model	Physical Activity, Level of PA	Satisfaction with Life Scale (SWLS), Satisfaction Level	Perceived Stress Scale (PSS-10), Stress Level
Risky	Insufficient	Low satisfaction	High
		Moderate satisfaction	
			Moderate
		High satisfaction	High
Acceptable	Sufficient	Moderate satisfaction	Low
	Increased		High
	High	Low satisfaction	Low
		High satisfaction	Moderate
		Moderate satisfaction	
Optimal	Increased	High satisfaction	Moderate
			Low
	High		Moderate
			Low
Intermediate form	All other level combinations of PA, SWLS & PSS-10		

**Table 9 ijerph-19-08629-t009:** Models of student behavior depending on the gender group (*n*, %; 95% CI).

Group	Gender	*N*	Behavior Patterns					
			Risky Behavior	Intermediate Form	Acceptable	Optimal Behavior		
**Medical students** [1]	Male	152	4, 2.6 (0.1–5.21)	115, 75.7 (68.8–82.5)	23, 15.1 (9.4–20.8)	10, 6.6 (2.6–10.5)	*χ^2^* = 5.2, *p* > 0.05	*χ^2^* = 178.7 *p*_[1, 2]_ < 0.001 *χ^2^* = 298 *p*_[1, 3]_ < 0.001 *χ^2^* = 44.1 *p*_[2, 3]_ < 0.001
	Female	465	4, 0.9 (0.1–1.7)	382, 82.2 (76.7–85.6)	50, 10.8 (7.9–13.6)	29, 6.2 (4.0–8.4)		
**Pedagogical students** [2]	Male	163	5, 3.1 (0.4–5.7)	77, 47.2 (39.6–54.9)	24, 14.7 (9.3–20.2)	57, 35.0 (27.7–42.3)	*χ^2^* = 0.4, *p* > 0.05	
	Female	574	14, 2.4 (1.2–3.7)	285, 49.7 (45.6–53.7)	83, 14.5 (11.6–17.3)	192, 33.4 (29.6–37.3)		
**Athletes** [3]	Male	208	0	65, 31.3 (25.0–37.6)	35, 16.8 (11.8–21.9)	108, 51.9 (45.1–58.7)	*χ^2^* = 1.1, *p* > 0.05	
	Female	207	0	70, 33.8 (27.4–40.3)	40, 19.3 (13.9–24.7)	97, 46.9 (40.1–53.7)		
**Total**	Male	523	9, 1.7 (0.6–2.8)	257, 49.1 (44.9–53.4)	82, 15.7 (12.6–18.8)	175, 33.5 (29.4–37.5)	*χ^2^* = 15.9, *p* < 0.05	
	Female	1246	18, 1.4 (0.8–2.1)	737, 59.1 (56.4–61.9)	173, 13.9 (12.0–15.8)	318, 25.5 (23.1–27.9)		

## Data Availability

The data that support the findings of this study are available on request from the corresponding author. The data are not publicly available due to privacy restrictions.

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
