# Peer review of "Physical Activity, Life Satisfaction, Stress Perception and Coping Strategies of University Students in Belarus during the COVID-19 Pandemic"

_ijerph, 2022, doi:10.3390/ijerph19148629_

Round 1
Reviewer 1 Report
After the corrections made, I consider that the manuscript should be accepted. I congratulate the authors for their work.
Reviewer 2 Report
Thanks to the authors' careful revision, I feel that the quality of the article has been greatly improved and I endorse it for publication.
This manuscript is a resubmission of an earlier submission. The following is a list of the peer review reports and author responses from that submission.
Round 1
Reviewer 1 Report
The paper has some scientific significance and the structure of the article is clear. However, the novelty and scientific contribution of the article are not fully emphasized in the current study, while its experimental design, research ideas, and literature review are slightly inadequate and need to be optimized.My detailed comments are as follows:
1) Introduction: The overview of the background to COVID-19 (the first three paragraphs) takes up a lot of space, is somewhat redundant and not streamlined enough; there is too little review of the level of physical activity, satisfaction with life, perceived stress, choice of coping strategies and the relationship between these indicators, and the logic of the relationship is not supported by the relevant literature; the novelty of the article is not sufficiently emphasized in the Introduction. Should this work fill in some gaps that could not be addressed in previous articles? The objectives of the study are somewhat mixed, and it is suggested to be streamlined.
2) Experimental Design: In the selection of subjects, only students with medical background, sports background and educational background are considered, but the reasons for selecting such subjects are not explained, but the conclusion and discussion mainly focus on it. I think the understanding of the motivation of the study is not clear, and this is not mentioned in the Introduction; the time span of the questionnaire collection is not indicated.
3)Discussion: The overall logic of the paragraph layout is not clear enough, could the first paragraph briefly describe the findings? There is not much research contribution in the discussion; "Due to the specifics of presenting information to the population about the epidemiological situation,pedagogical students and the majority of student-athletes refused to believe in what had happened, denied the reality, and rarely radically changed their lifestyle. " The inference in the discussion regarding the apparent stratification of PA levels in education/sports is not convincing enough and needs to be supported by references, or are there other reasons?
4)Add a separator for the numbers over
Recommendations
1) Introduction section.
â‘ It is suggested that the first and second paragraphs can be integrated into one paragraph to describe the current status of the new crown epidemic, and the content should be streamlined.
â‘¡ It is suggested that the content of indicators and corresponding references should be added.
â‘¢It is suggested to add research gaps based on previous related literature and emphasize the innovation point of the article.
â‘£The purpose of the study suggests that Aim 1 and Aim 2 are integrated together.
2) Experimental design.
â‘ In the selection of the research subjects it is suggested to add to the discussion of the reasons for the selection of the subjects.
â‘¡It is suggested to add the time frame of questionnaire collection.
3) Discussion section
â‘ It is suggested that the first paragraph briefly describe the results found and discuss the results of the data obtained in the study in the following paragraphs.
â‘¡The conclusion is suggested to add more specific and in-depth contributions and implications of the study.
Author Response
Response to Reviewer 1 Comments
Point 1: 1) Introduction section.
â‘ It is suggested that the first and second paragraphs can be integrated into one paragraph to describe the current status of the new crown epidemic, and the content should be streamlined.
Response â‘ : The reviewer's recommendation has been implemented. With gratitude, authors
Point 2: 1) Introduction section.
â‘¡ It is suggested that the content of indicators and corresponding references should be added.
Response â‘¡: The Reviewer's recommendations are accepted.. Content of indicators and corresponding references added in the text and marked in red.
Point 3: 1) Introduction section.
â‘¢It is suggested to add research gaps based on previous related literature and emphasize the innovation point of the article.
Response â‘¢: We have included additional information in the text of the article.
Point 4: 1) Introduction section.
â‘£The purpose of the study suggests that Aim 1 and Aim 2 are integrated together.
Response â‘£: The reviewer's recommendation was accepted with gratitude and implemented.
Point 5: 2) Experimental design.
â‘ In the selection of the research subjects it is suggested to add to the discussion of the reasons for the selection of the subjects.
Response â‘ : The reasons for selecting the subjects are added to the text of the article.
Point 6: 2) Experimental design.
â‘¡It is suggested to add the time frame of questionnaire collection.
Response â‘¡: The time frame for collecting the questionnaires has been added to the text of the article.
Point 7: 3) Discussion section
â‘ It is suggested that the first paragraph briefly describe the results found and discuss the results of the data obtained in the study in the following paragraphs.
Response â‘ : The Reviewer's recommendations were taken into account and incorporated into the text of the article.
Point 8: 3) Discussion section
â‘¡The conclusion is suggested to add more specific and in-depth contributions and implications of the study.
Response â‘¡: The "conclusions" section has been expanded. Practical recommendations are given in the section after "Conclusions".
Please see the attachment.

Reviewer 2 Report
Comments [IJERPH] Manuscript ID: ijerph-1760972
- The last two sentences of the abstract should be combined.
- The suggestions about increasing physical activity among university student are vague. Please offer more specific ones.
- What kind of improvement can be done about medical students’ work conditions and education program to reduce the stress of them?
Author Response
Please see the attachment.
Response to Reviewer 2 Comments
Point 1. The last two sentences of the abstract should be combined.
Response 1: We optimized the last sentences of the abstract with the reviewer's opinion in mind.
Point 2: The suggestions about increasing physical activity among university student are vague. Please offer more specific ones.
Response 2: Necessary additions were made to the " Conclusions" section.
Point 3: What kind of improvement can be done about medical students’ work conditions and education program to reduce the stress of them?
Response 3: Practical recommendations are specified in the section after the "Conclusions" section.

Reviewer 3 Report
The manuscript "Physical Activity, Life Satisfaction, Stress Perception and Cop- 2 ing Strategies of University Students in Belarus During the 3 Covid-19 Pandemic" its very interesting, its only convenient to spell check required, english language and style
Author Response
Please see the attachment!
Response to Reviewer 3 Comments
Point 1: The manuscript "Physical Activity, Life Satisfaction, Stress Perception and Cop- 2 ing Strategies of University Students in Belarus During the 3 Covid-19 Pandemic" its very interesting, its only convenient to spell check required, english language and style
Response 1: A spelling and style check of the English language has been performed. The text is consulted by a native English speaker. All recommendations are taken into account. The text is optimized.

Reviewer 4 Report
- The use of acronyms in the abstract is not recommended. I suggest that the authors give the full names of the questionnaires.
- I recommend adding the educational stage of the students in the aim of the study.
- It is advisable to add an example item to each of the questionnaires used. Similarly, the reliability (cronbach's alpha) obtained in each questionnaire should be added.
- The results are very good. The figures help to clarify the data obtained
The article in general seems to me to be very correct and appropriate for publication in the journal. I recommend the authors to make the small changes I have suggested to further improve the quality of the manuscript.
Author Response
Please see the attachment
Response to Reviewer 4 Comments
Point 1. The use of acronyms in the abstract is not recommended. I suggest that the authors give the full names of the questionnaires.
Response 1: The reviewer's opinion is taken into account. The abstract has been modified. The changes are typed in red in the text.
Point 2: I recommend adding the educational stage of the students in the aim of the study.
Response 2: Information that the respondents were full-time students from universities in Belarus has been added to the text of the article.
Point 3: It is advisable to add an example item to each of the questionnaires used. Similarly, the reliability (cronbach's alpha) obtained in each questionnaire should be added.
Response 3: We have provided some information to address reviewer's remark. Unfortunately, introducing point-by-point examples of the questions in the text would have lengthened the article considerably. We refer to the fact that the questionnaires are standardized and all questions have been presented in the references. The reliability index (Cronbach's alpha) is calculated and entered into the body of the text.
Point 4: The results are very good. The figures help to clarify the data obtained The article in general seems to me to be very correct and appropriate for publication in the journal. I recommend the authors to make the small changes I have suggested to further improve the quality of the manuscript.
Response 4: The reviewer's opinion is gratefully received and appreciated.
